# SiamDCFF: Dynamic Cascade Feature Fusion for Vision Tracking

**DOI:** 10.3390/s24144545

**Published:** 2024-07-13

**Authors:** Jinbo Lu, Na Wu, Shuo Hu

**Affiliations:** School of Electrical Engineering, Yanshan University, Qinhuangdao 066000, China; lujinbo@stumail.ysu.edu.cn (J.L.); hus@ysu.edu.cn (S.H.)

**Keywords:** object tracking, feature guidance, dynamic attention, Siamese network, dynamic cascade feature fusion

## Abstract

Establishing an accurate and robust feature fusion mechanism is key to enhancing the tracking performance of single-object trackers based on a Siamese network. However, the output features of the depth-wise cross-correlation feature fusion module in fully convolutional trackers based on Siamese networks cannot establish global dependencies on the feature maps of a search area. This paper proposes a dynamic cascade feature fusion (DCFF) module by introducing a local feature guidance (LFG) module and dynamic attention modules (DAMs) after the depth-wise cross-correlation module to enhance the global dependency modeling capability during the feature fusion process. In this paper, a set of verification experiments is designed to investigate whether establishing global dependencies for the features output by the depth-wise cross-correlation operation can significantly improve the performance of fully convolutional trackers based on a Siamese network, providing experimental support for rational design of the structure of a dynamic cascade feature fusion module. Secondly, we integrate the dynamic cascade feature fusion module into the tracking framework based on a Siamese network, propose SiamDCFF, and evaluate it using public datasets. Compared with the baseline model, SiamDCFF demonstrated significant improvements.

## 1. Introduction

Single-object tracking in computer vision involves selecting a specific object in the initial frame and tracking it across subsequent frames. Accurate feature matching between the features of a template and the features of a search area is critical to this task. Depth-wise cross-correlation is a common feature fusion method [1,2,3], but trackers using this strategy have hit a performance bottleneck.

SiamFC [4] approaches single-object tracking by formulating it as a correlation matching task. It uses the template feature map as the convolution kernel and convolves this kernel with the search region feature map to create a single-channel response map for estimating the center position of the tracked object. However, this cross-correlation strategy loses much information during implementation, particularly the feature information essential for regression boundaries. SiamRPN [5] utilizes a double-branch structure in the feature fusion process and retains the output feature map’s channel count, alleviating regression information loss. However, SiamRPN’s feature extraction network uses AlexNet [6], which has a limited capability to extract high-level semantic features. SiamRPN++ [1] introduces an effective spatial-aware sampling strategy to mitigate the impact of translation invariance caused by padding operations and employs the sophisticated backbone network ResNet50 [7]. It utilizes the fusion mode of SiamRPN [5] to conduct multi-scale feature fusion on the feature maps from the last three stages of ResNet50, and the cross-correlation fusion mode of SiamRPN has been improved to implement a depth-wise cross-correlation operation to optimize the computation efficiency. SiamCAR [3], proposed in the same year, no longer uses a two-branch feature fusion but directly concatenates the multi-stage output feature maps from the backbone in the channel dimension, followed by a depth-wise cross-correlation. All aforementioned trackers initiate the tracking process by extracting features through a Siamese network [8], followed by feature integration using a dedicated fusion subnetwork. This pipeline requires less computation, and the design of the feature extraction network is more independent of the feature fusion network.

Another feature fusion method involves integrating the feature extraction process with the feature fusion process. In this approach, the backbone network fuses features during the extraction process, as seen in models like OSTrack [9] and Mixformer [10]. In these works, the high-resolution search region and template region were serialized and directly input into the single-branch backbone network. This allowed interaction between the feature extraction process and the feature fusion process, enhancing the efficiency of both. However, these backbone networks typically consist of stacked transformers or their variants. Inputting serialized high-resolution image tokens directly into these networks results in significant computational demands, which are detrimental to the efficiency of both training and deployment.

For fully convolutional Siamese network-based trackers, the processes of feature extraction and feature fusion are explicitly separated. Feature fusion is primarily achieved through cross-correlation or depth-wise cross-correlation operations. This fusion method only focuses on the region within the convolution kernel, making it a local fusion method. The features after fusion do not establish global dependencies, and trackers based on fully convolutional networks may be affected by such locality, thereby limiting the expressiveness of their features. In response to this hypothesis, we design a set of verification experiments in this paper to explore whether establishing global dependencies for the features output from depth-wise cross-correlation operations can significantly improve the performance of fully convolutional trackers based on Siamese networks. The transformer structure, which possesses a robust ability to model global information, utilizes the attention mechanism to establish global dependencies among features. In our experiments, we required a controllable attention mechanism to establish different levels of global dependencies for features. The controllable attention module, likened to the “syringe” of global information, was designed with reference to DAT [11], and it controls the model’s ability to establish global dependencies for features by setting the number of reference points, aiming to achieve controlled injection.

It is further found that the current feature fusion strategy only utilizes either a local fusion strategy or a global fusion strategy, and this fusion mechanism cannot perform multi-scale fusion of features. In order to solve this problem, this paper proposes a cascaded feature fusion strategy that can be utilized for both local feature fusion and global dependency establishment and integrate it into the tracking framework based on a Siamese network, thereby proposing SiamDCFF.

This paper is specifically divided into two main parts. Firstly, a set of verification experiments is designed around the controllability of the dynamic attention module to investigate whether establishing global dependencies for the features output by the depth-wise cross-correlation operation can significantly improve the performance of the fully convolutional trackers based on a Siamese network, providing experimental support for rational design of the structure of the dynamic cascade feature fusion module. Secondly, we integrate the dynamic cascade feature fusion module into the tracking framework based on a Siamese network, proposing SiamDCFF, and evaluate it using public datasets. Compared with the baseline model, SiamDCFF demonstrated significant improvements.

The main contributions can be summarized as follows:We introduce a dynamic attention mechanism in this paper and propose a local feature guidance module. We then utilize the controllability of this dynamic attention structure to design comparative experiments, verifying that the performance of fully convolutional trackers based on a Siamese network can be significantly improved by establishing global dependencies for the features output from depth-wise cross-correlation operations.According to the results of comparative experiments, we present a dynamic cascade feature fusion module. This module cascades a depth-wise cross-correlation module, a local feature guidance module, and the dynamic attention modules to achieve multi-level integration among features. Furthermore, we propose a tracking model named SiamDCFF, which can model the dynamic process of local-to-global feature fusion.

## 2. Related Works

Here, we mainly review the work of other researchers in the fields of Siamese networks and vision transformers (VITs) [12], because the architectures used in these two areas are the main structures for visual tracking which have been used in recent years. Our model incorporates both architectures, and the relationship between them is also discussed in this paper to some extent.

### 2.1. Siamese Network-Based Visual Trackers

SiamFC [4] uses a Siamese network structure for the tracking model framework and frames the tracking problem as a similarity matching task, providing a well-defined framework for single-object tracking. To identify the object to be tracked, SiamFC requires two video frames as input: the first frame (template) and the subsequent search frame (search region). To maintain a strong correlation between these inputs, it is essential to use a backbone network with a consistent set of parameters for feature extraction. By directly convolving the template’s feature map with that of the search frame, SiamFC fuses features and produces a single-channel response map. However, this method has a low channel-to-information ratio and is not highly accurate in predicting object sizes. To enhance performance, SiamRPN [5] introduces a double-branch structure into the feature fusion process. Specifically, the template feature map, as output by the backbone network, is expanded to 2000 and 4000 times its original channel count using two convolution kernels. The convolution operation with the search feature map then yields multi-channel prediction results.

However, the backbone network of SiamRPN utilizes a shallow network, which limits its feature extraction capability. In 2018, Bo Li et al. proposed SiamRPN++ [1], which uses ResNet50 [7] as the backbone network to enhance feature extraction. To mitigate the damage caused by translation invariance from padding in deep convolutional neural networks, SiamRPN++ introduces a spatial awareness strategy. It successfully incorporates ResNet into the Siamese network framework. It employs a two-branch fusion strategy from SiamRPN for multi-scale feature fusion by combining feature maps across network stages. This deep CNN approach, while enhancing performance, increases the model’s parameter count, thus requiring careful hyperparameter management. Also in 2018, to reduce the need for excessive hyperparameter tuning, Guo et al. proposed the anchor-free SiamCAR model [3], which implements lightweight processing during the feature fusion stage. SiamCAR’s backbone network uses ResNet50, eliminating the double-branch feature fusion mode. Instead, it adopts a fusion approach which first concatenates multi-stage feature maps and then performs depth-wise cross-correlation. This approach not only boosts the model’s tracking speed but also maintains its accuracy. SiamCAR [3] is a typical fully convolutional tracker based on Siamese network. Owing to its more concise structure, SiamCAR [3] was adopted as the baseline model in this paper.

### 2.2. Vision Transformer

Initially introduced for sequence problems in natural language processing, the transformer [13] overcomes the parallelism limitations of RNNs and offers comprehensive global modeling. Its attention mechanism, featuring dynamic computation, enables the model to flexibly respond to varying inputs.

Prior to the vision transformer (VIT) [12], there have been efforts to integrate the attention mechanism into the tracking field. In 2020, SiamGAT [14] proposed modeling global information by replacing depth-wise cross-correlation feature fusion with graph attention. Also from that year, SiamAttn [15] introduces a dynamic Siamese attention module to enhance the spatial and channel dimensions of the multi-scale features output by the backbone network. In 2021, TrDiMP [16] integrated transformer encoders into template branches to strengthen the template features and used a decoder for feature fusion with search features, effectively integrating the transformer module directly into the Siamese network architecture and thereby replacing depth-wise cross-correlation operations with feature fusion. Following this, the VIT [12] constructs a vision transformer as the backbone network, focusing on processing images through convolutional layers and introducing the transformer into the field of image processing to serve as a feature extraction network. The VIT models images on a global scale and can effectively extract comprehensive global image information.

From the work in the two directions mentioned earlier, we observed that the fully convolutional trackers based on Siamese networks employ depth-wise cross-correlation operations for feature fusion. However, the features of depth-wise cross-correlation module outputs lack global dependencies, which may be a crucial factor in limiting the performance of fully convolutional trackers based on Siamese networks. Therefore, this paper first conducts comparative experiments to confirm the aforementioned hypothesis. We introduce a controllable global dependency modeling module into the feature fusion stage of the baseline model, SiamCAR [3], to construct multiple models with varying global dependency modeling capabilities. We then compare the performance of these models with SiamCAR [3]. Based on the results of these comparative experiments, we propose a new single-object tracking model by incorporating a dynamic cascade feature fusion module into a tracking framework based on Siamese networks.

## 3. Method

This chapter mainly introduces the proposed SiamDCFF model structure, with the detailed structure shown in Figure 1. The model comprises three main components: a backbone network based on a variant of ResNet50, a dynamic cascade feature fusion module, and a detection head designed for object location and regression.

### 3.1. Dynamic Attention Module

The global dependency modeling module used for comparative experiments should meet the following conditions. First, it should allow stacking to ensure adequate feature fusion. Second, any modifications to the module should not affect the original model structure to maintain the accuracy of the comparative experiments. Third, the module should allow control of the number of sampling points while maintaining a strong internal relationship among the sampled features. Traditional attentional mechanisms satisfy the first two criteria but do not satisfy the third. Therefore, this paper utilizes a dynamic attention mechanism as the feature global dependency modeling module.

As shown in Figure 2, the dynamic attention mechanism in this paper is based on the deformable transformer module in DAT [11], which was originally designed as the backbone network in the field of object detection. Its main purpose is to reduce the amount of computation. In this paper, the controllability of the dynamic attention mechanism is utilized in comparative experiments. Specifically, we resized the dynamic attention input to match the output feature map of the baseline model’s depth-wise cross-correlation module. Dynamic attention first set reference points according to the input feature map size, and we controlled the complexity by adjusting the number of reference points. The implementation process of the dynamic attention module is mainly divided into two steps. The first step involves subpixel-level sampling of key information in the feature map, which is the core of the dynamic attention mechanism and crucial for controlling its global modeling capability and computation cost. In the second step, the key sequence (*K*) and value sequence (*V*) are generated using the key information feature sequence obtained by sampling. Subsequently, the query sequence (*Q*) is generated using the unsampled feature map. Finally, standard self-attention operations are performed on the three previously mentioned sequences. The two steps mentioned above are described in the following sections:(1)$$\begin{array}{cc} \; & P=Grid\left(H_{g},W_{g},r\right),\;H_{g},W_{g}=\left\lfloor H/r\right\rfloor ,\left\lfloor W/r\right\rfloor ,\\ \delta = & s\times \tanh\left(\theta_{offset}\left(Q\right)\right)=s\times \tanh\left(Conv\left(\;XW^{q}\;\right)\right),\\ \; & \;\;\;\;\tilde{X}=Bilinear\left(X;P+\delta \right). \end{array}$$
where Grid represents the grid generation function, which is used to generate a grid with *H* rows and *W* columns on the feature map according to the stride *r*; δ represents the generated offset map, the size of which is dependent on the sampling point setting; *P* represents the reference point coordinates; $\tilde{X}$ represents the sampled feature sequence; and Bilinear represents the bilinear sampling function. The specific calculation process is presented in Equation (Equation 2).

Since the offset generated is generally floating-point data, and considering that the image is structured with integer pixels as the unit of measurement, we were prompted to use bilinear interpolation to extract features from the feature map. The procedure is as follows:(2)$$\begin{array}{c} Bilinear\left(f,\left(P_{\delta_{i}},P_{\delta_{j}}\right)\right)=\sum_{\left(I_{x},I_{y}\right)}m(|P_{\delta_{i}}-I_{x}\left|\right)\\ m(|P_{\delta_{j}}-I_{y}\left|\right)f\left[I_{y},I_{x},:\right]. \end{array}$$
where *f* represents the feature map for sampling; $\left(P_{\delta_{i}},P_{\delta_{j}}\right)$ represents the coordinates of the sampling points; and m(a)=max(0,|a|). $\left(I_{x},I_{y}\right)$ represents the index of all pixel points on *f*. Each item added in the above equation represents the weighted addition of the four points closest to the sampling point on the original feature map, which we take as the sample value.

After adding the offset to the reference points set earlier, we performed the sampling operation on the original feature map and then resized $K^{\prime }$ and $V^{\prime }$ according to the results obtained from bilinear transformation of the sampling points. This is the first screening of *K* and *V*, which ensures that the screening features are relevant to the tracked object. In essence, this involves correlation preprocessing, which also helps control the computation cost, enabling this module to serve as the backbone network component or the feature fusion component. *Q* plays a guiding role in the calculation process and contains guiding information, and thus sample processing was not performed. Therefore, it was equivalent to only processing *K* and *V*. In this way, there was an additional advantage in controlling the scale of modification to prevent unnecessary interference. For the offset generation process, it can be considered a trainable offset adaptive sampling strategy. This allowed us to extract features which had a higher correlation with *Q* through training. Compared with only using reference points for feature sampling, this method preserves the internal relationship between features and offers higher reliability. After obtaining $K^{\prime }$ and $V^{\prime }$, we calculated the attention between *Q* and $K^{\prime }$. Compared with the method of extracting *K* points with the highest attention value after the attention operation of *Q* and *K*, this approach can significantly reduce the computation cost while preserving performance to the fullest extent possible. In this paper, we utilize dynamic attention in the feature fusion module to dynamically establish global feature dependencies. The specific execution process is illustrated in the following formula:(3)$$\begin{array}{cc} \; & Q=XW^{q},K^{\prime }=\tilde{X}W^{k},V^{\prime }=\tilde{X}W^{v},\\ \; & A^{\left(m\right)}=softmax\left(\frac{Q^{T(m)}{K\prime }^{\left(m\right)}}{\sqrt{d}}\right),\\ Attn(Q, & K^{\prime },V^{\prime })=Concat\left(A^{\left(1\right)}{V\prime }^{\left(1\right)},...,A^{\left(n\right)}{V\prime }^{\left(n\right)}\right)W^{o}. \end{array}$$
where $W^{q}\in R^{C_{in}\times C_{dim}}$, $W^{k}\in R^{C_{in}\times C_{dim}}$, and $W^{v}\in R^{C_{in}\times C_{dim}}$ are three linear projection layers; *d* is the dimension of *Q*; and $W^{o}\in R^{C_{dim}\times C_{out}}$ is used for further semantic extraction.

### 3.2. Location for Adding Global Information

For the location of the introduction of the global dependency modeling module, this paper presents two requirements for designing the model. (1) We needed to incorporate the global information fusion module into the basic depth-wise cross-correlation fusion strategy utilized in the baseline model while keeping the other components unchanged. We could not modify existing parts of the baseline model; (2) We did not modify configuration hyperparameters such as the amount of training data used by the models but simply added global modules to the basic existing structures. In addition, for implementation of the comparison experiments, we needed to stabilize the model structure. If the feature extraction network were replaced with a transformer, then the experimental configuration had to align with the transformer structure because of the significant changes in experimental parameters, which would violate the second requirement mentioned earlier. The transformer was positioned directly behind the detection head to encode the response maps. This process essentially re-encodes the decoded information, which contradicts our experimental purpose and lacks a scientific model structure. If the global feature fusion module is placed before the depth-wise cross-correlation operation, the feature map still needs to undergo local fusion in the end. This results in a local induction bias of the features input into the detection subnetwork. After analysis, to preserve the baseline model structure and components, and considering that the dynamic attention module could only accept one input, the most suitable placement was after the depth-wise cross-correlation feature fusion and before the detection subnetwork.

### 3.3. Local Feature Guidance Module

Since this paper aims to demonstrate that the performance of fully convolutional trackers based on a Siamese network can be significantly improved by establishing global dependencies for the features output from depth-wise cross-correlation operations, it was necessary to incorporate a global dependency modeling capability into the model. However, it may not be appropriate to directly input the feature map after the depth-wise cross-correlation module fusion process into the dynamic attention module. Since the feature map generated by depth-wise cross-correlation involves structured matching, which differs from the serialized detail fusion of the attention mechanism, an intermediate process is required to integrate the dynamic attention module. To achieve this, we proposed a local feature guidance module implemented by window attention to precisely match local details. In addition, the window attention module can enhance the efficiency of injecting global information into the subsequent single-layer dynamic attention module. The window attention module serves as a transitional operation between local information fusion and global information fusion, functioning as a preprocessing operation in this context. Compared with adding an extra layer of a vanilla transformer, the computation cost could be significantly reduced. The following section describes the execution process of the transition phase.

After the depth-wise cross-correlation, the spatial dimension of the generated feature map was 25×25. We divided the fused feature map into 25 5×5 windows, and then we performed standard attention operations in each window. Subsequently, an MLP module was used to further extract the features. The complete calculation process in each window is as follows:(4)$$\begin{array}{cc} \; & Q_{w}=X_{w}W^{q},K_{w}=X_{w}W^{k},V_{w}=X_{w}W^{v},\\ \; & A_{w}^{\left(m\right)}=softmax\left(\frac{Q_{w}^{T(m)}K_{w}^{\left(m\right)}}{\sqrt{d}}\right),\\ Attn_{w}(Q_{w}, & K_{w},V_{w})=Concat\left(A_{w}^{\left(1\right)}V_{w}^{\prime (1)},...,A_{w}^{\left(n\right)}V_{w}^{\prime (n)}\right)W^{o}. \end{array}$$
where $X_{w}$ is the sequence of features inside the window abd $Attn_{w}$ is the window attention operation.

### 3.4. Dynamic Cascade Feature Fusion Module

According to the comparative experimental results in Section 4.3, we cascaded a depth-wise cross-correlation module, a local feature guidance module, and dynamic attention modules to create a dynamic cascade feature fusion (DCFF) module. First, the depth-wise cross-correlation module received three pairs of feature maps from the network output of the feature extraction subnetwork. It concatenated multi-scale feature maps according to the channel dimensions, called *Z* and *X*, and then clipped the 7×7 region in the center of *Z*. It then obtained a template convolution kernel $Z_{crop}\in R^{7\times 7\times (256\times 3)}$ and conducted a depth-wise cross-correlation operation between $Z_{crop}$ and *X* to generate a fusion feature map with dimensions of 25×25×768. After that, the fusion feature map was divided into 25 windows (5×5), and standard self-attention operations were performed in each window to further establish the precise local dependencies among the features. Finally, the feature map output by the local feature guidance module was input into the dynamic attention modules. A certain number of features with strong expressiveness for tracking an object’s spatial information were sampled on the feature map, based on the reference points and offset. These features were used to generate the $K\in R^{B\times 768\times n}$ sequence and $V\in R^{B\times 768\times n}$ sequence. The unsampled feature map was then used to generate the $Q\in R^{B\times 768\times n}$ sequence. The three sequences above were executed with an attention operation to accomplish dynamic cascade feature fusion. After the feature fusion was completed, the fused feature map was passed through a deconvolution layer with 768 input channels and 256 output channels to generate the output of the feature fusion stage.

### 3.5. Detection Subnetwork

The detection subnetwork primarily performs two tasks; one is classification, which involves distinguishing between the foreground and background of the image, and the other is regression, which determines the position of the tracked object and establishes the rectangular boundaries between the background and the tracked object.

The detection subnetwork utilized in this paper was identical to the one in SiamCAR [3], and the primary structure of the classification branch mirrored that of the regression branch. Each stack consisted of four convolutional blocks, where each convolutional block comprised a convolutional layer with 256 input channels and 256 output channels, a group normalization layer, and a ReLU activation function. For the classification branch, following the body of the classification branch, there was a 3×3 convolutional kernel with 2 output channels. This kernel was used to generate foreground and background classification scores for each position on the feature map. In addition, after the body of the classification branch, there was a 3×3 convolutional kernel with 1 output channel which generated the center-ness score to constrain the corresponding classification map. For the regression branch, its body was followed by a convolutional kernel with 4 output channels, which was used to determine the bounding box for the tracked object in regression:(5)$$\begin{array}{cc} \; & CGR(X)=Conv(GN(ReLU(X))),\\ \; & task_{i}\left(X\right)=Conv_{3\times 3}\left(CGR_{\times 4}\left(X\right)\right). \end{array}$$

Here, CGR represents the convolution block, while $task_{i}$ involves the classification branch, center-ness branch, and regression branch.

## 4. Experiments

The experiments in this paper were divided into two parts. The first part aimed to validate that the performance of fully convolutional trackers based on a Siamese network could be significantly improved by establishing global dependencies for the features output from depth-wise cross-correlation operations. This was achieved by controlling the models’ ability to establish global dependencies for features using the dynamic attention module, the details of which are shown in Section 4.3. The other part involved comparing the evaluation results of the proposed model with those of other trackers of the same type in public datasets, the detailed evaluation results of which are presented in Section 4.4.

### 4.1. Implementation Details

For the backbone network of the model, we adopted a variant of ResNet50, modifying the partial convolution layers of the last three stages to become dilated convolutions and changing the strides of the last three stages to 1. This adjustment ensured that the output feature size remained consistent, facilitating smooth execution of the subsequent feature fusion process. We trained all models in the comparison experiments using a server with two 1080Ti GPUs. In order to control the variables, both our model and the baseline model were initialized using backbone network weights pretrained on the ImageNet dataset to expedite model convergence. The training datasets used in the training process are all public datasets, including COCO [17], ImageNet VID [18], YOUTUBEBB [19], and ImageNet DET [18]. The number of training epochs was set to 20.

In the training process, we set the number of images to be learned in each epoch to 600,000, using the repeatability principle to maintain this data amount throughout. There were 20 epochs of training. During the initial phase of training, the backbone network parameters of the model were frozen for the first 10 epochs. Starting from the 11th epoch, the parameters of the last three stages of the backbone network were unfrozen and included in the training. The rest of the configuration remained unchanged from the baseline SiamCAR [3] model.

### 4.2. Dataset Introduction

This paper uses the following evaluation datasets:

**UAV123**: UAV123 [20], a quintessential long-term tracking dataset with 123 video sequences, encompasses 12 prevalent challenges, like blurring and fast movement. UAV123 follows the OPE evaluation strategy, ensuring a scientific evaluation process. UAV123 uses precision and success as evaluation metrics, in which precise and normalized precision (Norm.P) measure the distance between the center of the predicted bounding boxes and the center of the ground truth boxes. The success measure calculates the intersection over union (IOU) between the predicted bounding boxes and the ground truth boxes.

**GOT-10k**: GOT-10k [21] is a large dataset containing over 10,000 video sequences for single-object tracking. It is an online evaluation dataset where all models are consistently sent to the background server for evaluation, ensuring fairness in the evaluation process. GOT-10k is an authoritative dataset in the field of single-object tracking. GOT-10k utilizes the average overlap (AO) and success rate (SR) as evaluation metrics. The average overlap measures the mean of the IOU between the predicted bounding boxes and the ground truth boxes for all categories. The success rate is defined as the proportion of frames where the IOU between the predicted bounding boxes and the ground truth boxes exceeds a specific threshold, relative to the total number of frames. The thresholds are usually set to 0.5 and 0.75, respectively. The frames per second (FPS) measure indicates how many frames per second can be tracked.

**OTB100**: The OTB100 [22] dataset contains 100 challenging video sequences and is a public evaluation dataset. It presents 11 specific challenges for evaluating the tracker, including fast movement and a low resolution. OTB100 complies with the OPE evaluation strategy, ensuring a scientific evaluation process. OTB100 also uses precision and success as evaluation metrics, which have the same meanings as in UAV123.

### 4.3. Comparative Experiments

According to the discussion in Section 2.1 on the current fully convolutional trackers based on a Siamese network, we can conclude that SiamCAR [3] is a typical fully convolutional tracking model, owing to its more concise structure. Its structure can represent fully convolutional trackers based on Siamese networks well. For our comparison experiments, we selected SiamCAR as the baseline model. In this paper, five model structures with varying capabilities in global dependency modeling are designed based on SiamCAR [3]. The first model is SiamCAR + L, which adds a local feature guidance module after the depth-wise cross-correlation operation of SiamCAR [3]. The second model is SiamCAR + L + 1/2D, which adds a dynamic attention module layer with the number of sampling points set to (H/2)×(W/2) at the back of the local feature guidance module in SiamCAR+L. The third model is SiamCAR+L+D, which adds a dynamic attention module layer with the number of sampling points set to *H*×*W* to the back of the local feature guidance module in SiamCAR+L. The fourth model is SiamCAR+L+3D, which adds three layers of dynamic attention modules with the number of sampling points set to *H*×*W* to the back of the local feature guidance module in SiamCAR+L. The fifth model is SiamCAR+L+6D, which adds 6 dynamic attention module layers with the number of sampling points set to *H*×*W* to the back of the local feature guidance module in SiamCAR+L. For these five models, we aimed to minimize modifications to other parts to prevent any interference with the experimental results. Additionally, we needed to exclude the effects of irrelevant variables. We also trained the baseline model in the same environment and named it SiamCAR-Ours. Other original configurations and training strategies aligned with this paper. For the training, the dataset used was the same as the one specified in the experimental details. Then, the experimental results of our modified model using the UAV123 and GOT-10k evaluation indices were compared with those of other trackers. The evaluation results are shown in Table 1. The results show that the tracking effect of the model improved after incorporating global dependencies into the feature fusion stage compared with the baseline model. The hypothesis regarding the lack of a global dependency modeling ability in fully convolutional tracking models has been validated, and the effectiveness of the dynamic cascade feature fusion module has been demonstrated.

As shown in Table 1, all models were tested under the same experimental conditions (SiamCAR_paper represents the results of this paper). In comparison with SiamCAR_paper, SiamCAR+L+1D showed a 2.7% improvement in terms of Norm.P. SiamCAR+L+1D also showed a 1.9% improvement in terms of the success rate. More significantly, compared with SiamCAR_Ours, in terms of precision, our model achieved a 4.3% improvement. In terms of Norm.P and success, the improvements were 3.4% and 2.6%, respectively. It can be seen that the model’s performance significantly improved after incorporating global information compared with the baseline model.

As shown in Table 2, similar experimental results to those in Table 1 were also achieved on the GOT-10k dataset (SiamCAR_paper represents the results of this paper). Compared with the evaluation results of the baseline model under the same experimental conditions, SiamCAR+L+3D showed improvements of 2.0% in terms of AO and 2.7% in terms of SR_0.5_ over the baseline model. Additionally, it improved by 4.8% in terms of SR_0.75_.

It can be seen in Table 1 and Table 2 above that on the UAV123 dataset, SiamCAR+L+1D outperformed SiamCAR+L+3D. Conversely, on the GOT-10k dataset, SiamCAR+L+3D demonstrated superior performance to SiamCAR+L+1D. This discrepancy can be attributed to the dynamic attention module’s dual role; not only does it establish global dependencies among features, but it also possesses a filtering capacity. On short-term tracking datasets such as GOT-10k, the filtering effect was less pronounced. But on long-term tracking datasets like UAV123, where the tracked object’s state could have more pronounced changes over the tracking period and as the dynamic attention modules were stacked, this filtering capability became stronger. This filtering ability might inadvertently discard some valuable information, but it is trainable, and its accuracy improved with the increase in training data. It can also be observed from the tables that only adding a local guidance module to the baseline model (SiamCAR+L) resulted in a decline in model performance. This was due to the fact that the size of the local window was manually defined, and non-overlapping windows can lead to a lack of information interaction. Therefore, performance decreased when relying solely on the local feature guidance module. After incorporating a dynamic attention module, the issue of non-interaction between different windows of the local guidance module was addressed, and this could be viewed as a transitional processing step. The significant improvement in the results of SiamCAR+L+1/2D in the tables also substantiates this point. In addition, the tables indicate that stacking dynamic attention modules to six layers did not improve performance. In fact, the performance deteriorated to some extent. We suggest that this phenomenon occurred because when stacking six layers of dynamic attention modules, the global information needed for the feature fusion stage in this configuration became saturated. When stacking dynamic attention modules continuously, the increased number of parameters did not correspond to the actual amount of training data, leading to a certain degree of overfitting. In addition, the comparative experiments also demonstrate that the enhancement of the model’s effectiveness was not due to the experimental conditions. Even under identical conditions, there was a significantly better performance improvement compared with the baseline performance.

In the above comparative experiments, SiamCAR+L+1D delivered the top performance on the UAV123 benchmark. It also attained results comparable to the first place on the GOT-10k dataset and significantly outperformed the baseline model. Consequently, the experimental results confirm that the performance of fully convolutional trackers based on a Siamese network can be substantially enhanced by establishing global dependencies for the features derived from depth-wise cross-correlation operations.

### 4.4. Evaluation on Public Datasets

Based on the analysis of the comparative experimental results, and considering the computation cost, this paper set the number of layers of the dynamic attention module in the dynamic cascade feature fusion module to one, and a new tracking model was constructed using this module, named SiamDCFF. However, we found that directly applying the post-processing strategy of the baseline model to SiamDCFF was not the optimal choice. Therefore, this paper modified the post-processing process used by SiamDCFF to enhance the regression accuracy. Specifically, we used the prediction results of the current frame directly as the final regression result and fine-tuned the penalty coefficient which was applied to the classification response map. Finally, SiamDCFF was evaluated on the UAV123, GOT-10k, and OTB100 benchmarks.

**Evaluation Results on UAV123:** Compared with other similar trackers in Table 3, in terms of success, our model was 0.7% lower than the first-place TrTr-online [23] and 0.5% lower than the second-place SiamAttn [15]. In terms of Norm.P, our model was 0.6% lower than the first-place SiamGAT [14] and 1.2% higher than the third-place Ocean [24]. In terms of precision, our model tied for first place with SiamAttn and was 1.0% higher than the third-place Ocean [24]. Compared with the model without dynamic attention (i.e., the model structure of SiamCAR [3]), the results improved even more when global information was added. As can be seen from Table 3, compared with SiamCAR [3], in terms of success, our model’s result was higher by 3.1%, and in terms of Norm.P, our model’s result was higher by 4.0%.

**Evaluation Results on GOT-10K:** The evaluation results on GOT-10K are shown in Figure 3. Compared with other trackers of the same type in Table 4, in terms of AO, the model proposed in this chapter was 0.1% lower than DiMP50 [28], which ranked first, and ranked third was DCFST [29]. In terms of SR_0.5_, it was 0.1% higher than the second-ranked DiMP50 and 0.2% higher than the third-ranked DCFST. In terms of SR_0.75_, it was 0.2% higher than the second-ranked DiMP50 and the third-ranked DMV50 [30]. Compared with the original Siamese structure model SiamCAR [3], our experimental results showed significant improvement. Specifically, there was a 4.1% increase in terms of AO and a 4.8% increase in terms of SR_0.5_, while in terms of SR_0.75_, there was a 7.9% improvement.

**Evaluation Results on OTB100:** Compared with the other trackers in Table 5, in terms of precision, our model was 1.7% lower than TrTr-online [23] and 0.4% lower than Ocean-online [24]. In terms of the success rate, our model was 1.1% lower than TrTr-online [23] but was 0.8% higher compared with both SiamBAN [2] and SiamRPN++ [1]. Additionally, it was 2% higher compared with Ocean-online. Compared with the model without dynamic attention (i.e., the model structure of SiamCAR [3]), the results improved even more when we added global information. As can be seen in Table 5, compared with SiamCAR, in terms of precision, our model was 0.6% higher, and in terms of success rate, our model was 0.7% higher.

### 4.5. Visualization of Tracking Results

A visualization of the tracking results is shown in Figure 4. The model proposed in this paper could maintain relatively stable tracking results in the following three representative video sequences. Compared with the baseline model, our model had significantly improved tracking performance.

## 5. Conclusions

We verified through experiments that creating global dependencies for features can enhance model performance. This also confirms the observation that, in general, the performance of fully convolutional trackers based on Siamese networks can be significantly improved by establishing global dependencies for the features output from depth-wise cross-correlation operations. In addition, we demonstrated that dynamic attention modules can be effectively introduced into single-object tracking tasks. The next step is to further optimize the structure of the dynamic cascade feature fusion module and evaluate its optimal performance. Furthermore, we need to integrate the local feature fusion process with the global feature fusion process to enhance the efficiency of feature fusion.

## Figures and Tables

**Figure 1 sensors-24-04545-f001:**
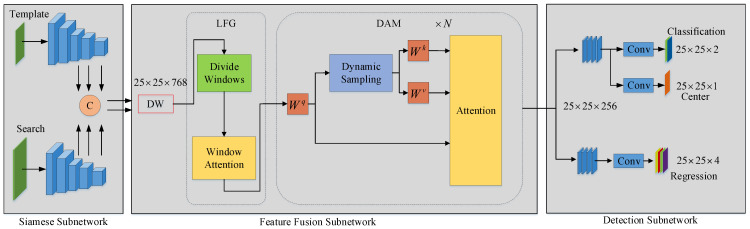
This is an overview of the structure of the proposed model, including the Siamese subnetwork, feature fusion subnetwork, and detection subnetwork. Here, C represents the concat of the channel dimension; DW represents the depth-wise cross-correlation operation; LFG represents the local feature guidance module (Section 3.3); DAM stands for the dynamic attention module (Section 3.1); and ×N represents the stacked layers of the DAM.

**Figure 2 sensors-24-04545-f002:**
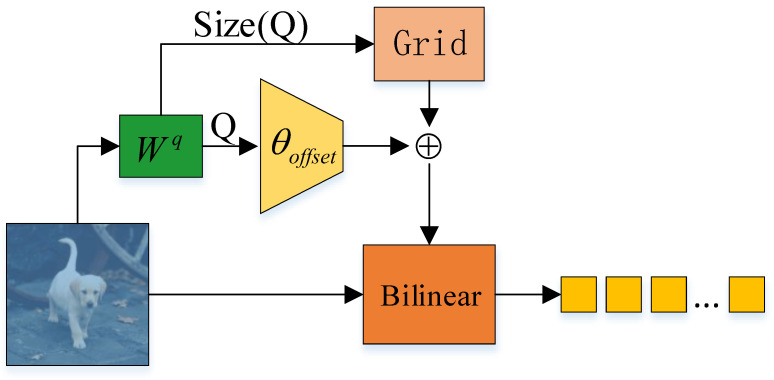
An illustration of the dynamic sampling process. This sampling strategy yields sampling points which have intrinsic connections.

**Figure 3 sensors-24-04545-f003:**
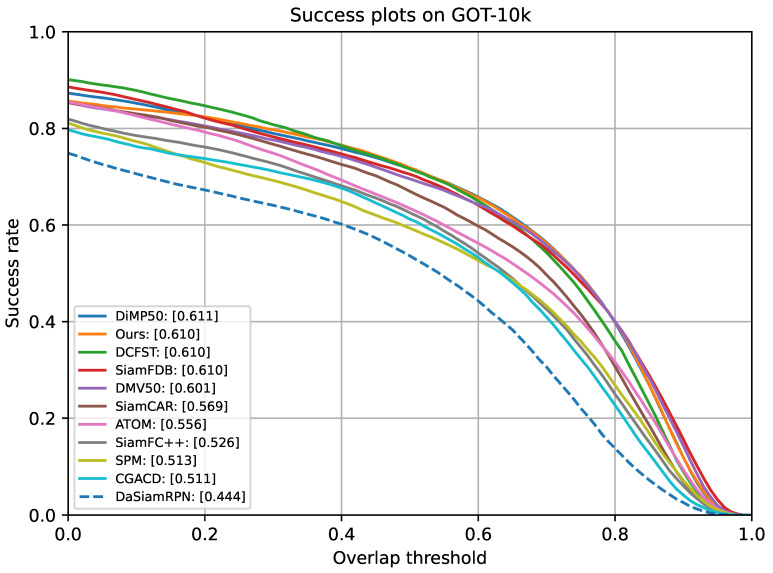
Success plot on GOT-10K [21].

**Figure 4 sensors-24-04545-f004:**
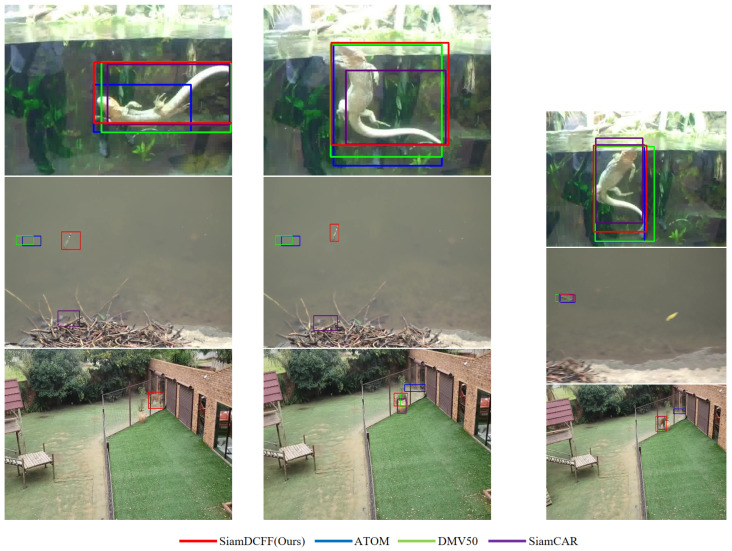
Our SiamDCFF model compared with three analogous trackers using representative sequences on the GOT-10K [21] dataset, showing that our model outperformed the others by accurately predicting bounding boxes under challenging conditions, including aspect ratio changes, fast motion, and partial occlusion. Meanwhile, SiamCAR [3] and DMV50 [30] produced far rougher results. In addition, our model’s predictive results differed little from the artificially labeled bounding boxes of the tracked objects in the above cases.

**Table 1 sensors-24-04545-t001:** The results of comparative experiments on UAV123 [20] benchmark.

Tracker	Success Rate ↑	Norm.P ↑	Precision ↑
SiamCAR_paper [3]	0.614	0.760	-
SiamCAR_Ours	0.607	0.753	0.788
SiamCAR+L	0.621	0.773	0.811
SiamCAR+L+1/2D	0.623	0.773	0.818
SiamCAR+L+1D	**0.633**	**0.787**	**0.831**
SiamCAR+L+3D	0.623	0.779	0.815
SiamCAR+L+6D	0.610	0.759	0.795

**Table 2 sensors-24-04545-t002:** The results of comparative experiments on GOT-10K [21] benchmark.

Tracker	AO ↑	SR_0.5_ ↑	SR_0.75_ ↑
SiamCAR_paper [3]	0.569	0.670	0.415
SiamCAR_Ours	0.547	0.635	0.426
SiamCAR+L	0.541	0.608	0.409
SiamCAR+L+1/2D	0.584	0.687	**0.467**
SiamCAR+L+1D	0.586	0.688	0.455
SiamCAR+L+3D	**0.589**	**0.697**	0.463
SiamCAR+L+6D	0.552	0.640	0.434

**Table 3 sensors-24-04545-t003:** The evaluation on the UAV123 [20] benchmark.

Tracker	Success ↑	Norm.P ↑	Precision ↑
SiamDW [25]	0.536	0.728	0.776
SiamRPN [5]	0.581	0.737	0.772
HiFT [26]	0.589	0.758	0.787
TrTr-offline [23]	0.594	-	-
SiamRPN++ [1]	0.611	0.765	0.804
SiamCAR [3]	0.614	0.760	-
CGACD [27]	0.620	0.782	0.815
Ocean [24]	0.621	0.788	0.823
SiamBAN [2]	0.631	-	0.833
SiamGAT [14]	0.640	0.806	0.835
SiamAttn [15]	0.650	-	0.845
TrTr-online [23]	0.652	-	-
**Ours**	0.645	0.800	0.845

Here, the colors red, green, and blue denote the sequence of indications, ranked from first to third, respectively.

**Table 4 sensors-24-04545-t004:** The evaluation on the GOT-10k [21] benchmark.

Tracker	AO ↑	SR_0.5_ ↑	SR_0.75_ ↑	FPS
DaSiamRPN [31]	0.444	0.536	0.202	134.40
CGACD [27]	0.511	0.612	0.323	37.73
SPM [32]	0.513	0.593	0.359	72.30
SiamFC++ [33]	0.526	0.625	0.347	186.29
ATOM [34]	0.556	0.634	0.402	20.71
SiamCAR [3]	0.569	0.670	0.415	17.21
SiamFDB [35]	0.586	0.658	0.434	36.52
DMV50 [30]	0.601	0.695	0.492	31.55
DCFST [29]	0.610	0.716	0.463	50.65
DiMP50 [28]	0.611	0.717	0.492	34.05
**Ours**	0.610	0.718	0.494	21.76

Here, the colors red, green, and blue denote the sequence of indications, ranked from first to third, respectively.

**Table 5 sensors-24-04545-t005:** The evaluation on the OTB100 [22] benchmark.

Tracker	Precision ↑	Success ↑
GCT [36]	0.853	0.647
ECO-HC [37]	0.856	0.643
DaSiamRPN [31]	0.880	0.658
TrTr-online [23]	0.901	0.691
Ocean-offline [24]	0.902	0.673
PrDiMP [38]	-	0.696
SiamCAR [3]	0.910	0.697
SiamBAN [2]	0.910	0.696
SiamRPN++ [1]	0.915	0.696
Ocean-online [24]	0.920	0.684
TrTr-online [23]	0.933	0.715
**Ours**	0.916	0.704

Here, the colors red, green, and blue denote the sequence of indications, ranked from first to third, respectively.

## Data Availability

Data are contained within the article.

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
