# Peer review of "SiamDCFF: Dynamic Cascade Feature Fusion for Vision Tracking"

_sensors, 2024, doi:10.3390/s24144545_

Round 1
Reviewer 1 Report
Comments and Suggestions for Authors
The author needs to explain about how does the proposed dynamic cascade feature fusion approach work?
Author need to tell what are the key components of the dynamic cascade feature fusion framework?
Author need to explain how does the proposed method improve upon existing feature fusion techniques in vision tracking?
Author need to tell what datasets and evaluation metrics are used to assess the performance of the proposed approach?
Reviewer 2 Report
Comments and Suggestions for Authors
The paper proposes a dynamic cascade feature fusion module (DCFF) by introducing a local feature guidance module (LFG) and dynamic attention modules (DAM) , which can enhance the global dependency modeling capability during the feature. My detailed comments are listed below, which hopefully can help the authors improve the quality of their work.
1. The novelty and motivations of this paper is not well stated. It is suggested to improve those parts.
2. In the Section 1 and 2 , the analysis of newly developed related technologies in recent years are insufficient, and need to be supplemented.
3. The comparison algorithm in the experiment is relatively old, with only one algorithm from 2023. Suggest adding some results published in 2023 or 2024 for comparison to highlight the superiority of the manuscript.
4. In addition to comparing objective evaluation indicators, the author should also add comparisons in terms of computational complexity, training time, and testing time.
5. Increased the related ablation study to demonstrate the effectiveness of the proposed modules.
Comments on the Quality of English Language
Minor editing of English language required.
Reviewer 3 Report
Comments and Suggestions for Authors
Lu et al. proposed a single-target tracking algorithm based on improved SiamCAR, which is innovative in that it proposes a Dynamic Cascade Feature Fusion (DCFF) module by introducing a Local Feature Guidance (LFG) module and Dynamic Attention Modules (DAM) after the depth-wise cross-correlation module. The algorithm is trained with a publicly available dataset and validated. However, in order for the paper to be at a publishable level, refinements need to be made in the following places:
1. The abstract section, lines 8 to 16 is an overview of the main content of the paper and it is suggested that it should be placed at the end of the introduction section.
2. Figure 1 is a visual comparison of the network of this paper with other networks, this image is suggested to be placed at the end of the discussion section, in addition please show multiple representative images.
3. Line 43 for the first time a twin network appears in the text and it needs to be referenced.
4. In lines 72-74, you said that you first conducted a comparison experiment, and then obtained a dynamic cascade feature fusion module based on the results of the comparison experiment, and embedded it into the twin network-based target tracking framework. I think this sentence is a bit inappropriate, the comparison experiment you mentioned is actually based on the SiamCAR framework, and gradually increase the modules you proposed to get an optimal combination of modules, I think this should be more properly called an ablation experiment, and what is in section 4.4 is more appropriately called a comparison experiment. This sentence appears several times in the text, so please check and revise it in full.
5. Please describe the meaning of the evaluation metrics in Table 1 and Table 2, and write their full names when they first appear.
6. Are the three datasets UAV123, GOT-10k, and OTB100 with different evaluation metrics? And only one success plots is shown, please add the success plots under the other two datasets.
7. Table 5 shows the evaluation on OTB100 benchmark. But your algorithm in this paper is lower than TrTr-online in terms of Precision and Success, please explain the reason specifically and clarify what is the advantage of your algorithm.
Comments on the Quality of English Language
Moderate editing of English language required.
Round 2
Reviewer 2 Report
Comments and Suggestions for Authors
I have carefully read the response of the authors to the comments and suggestions, and have no more questions.
Comments on the Quality of English Language
Minor editing of English language required.
Reviewer 3 Report
Comments and Suggestions for Authors
The paper can be accepted in its present form.
Comments on the Quality of English Language
Minor editing of English language required.